# Software-Defined Radio Implementation and Performance Evaluation of Frequency-Modulated Antipodal Chaos Shift Keying Communication System †

Arturs Aboltins * and Nikolajs Tihomorskis

Institute of Microwave Engineering and Electronics, Riga Technical University, 12 Azenes Street, LV-1048 Riga, Latvia
* Correspondence: aboltins@rtu.lv; Tel.: +371-26557340
† This Paper Is an Extended Version of Our Paper Published in Conference Proceedings of IEEE Wireless Communications and Networking Conference, WCNC 2023.

**Abstract:** This paper is devoted to software-defined radio (SDR) implementation of frequency modulated antipodal chaos shift keying (FM-ACSK) transceiver and presents results of prototype testing in real conditions. This novel and perspective class of spread-spectrum communication systems employs chaotic synchronization for the acquisition and tracking of the analog chaotic spreading code and does not need resource-demanding cross-correlation. The main motivation of the given work is to assess the performance of FM-ACSK in real conditions and demonstrate that chaotic synchronization can be considered an efficient spread-spectrum demodulation method. The work focuses on the real-time implementation aspects of the modulation-demodulation algorithms, forward error correction (FEC) and symbol timing synchronization approach in MATLAB Simulink. The performance of the presented prototype is assessed via extensive testing, which includes measurement of bit error ratio (BER) in single-user and multi-user scenarios, estimation of carrier frequency offset (CFO) impact and image transmission over-the-air between two independent sites and comparison with classical frequency hopping spread spectrum (FHSS). The paper shows that the presented class of the spread spectrum communication systems demonstrates good performance in low signal-to-noise ratio (SNR) conditions and in terms of BER significantly outperforms the classic spread-spectrum modulation schemes which employ correlation-based detection.

**Keywords:** software-defined radio; chaotic communication; spread spectrum communication; frequency modulation; synchronization; communication systems; MATLAB; physical layer

## 1. Introduction

Chaos-based digital communication systems have been researched for several decades. The development of the first electronic circuits capable of generating chaotic oscillations in the 90s raised a huge interest in the scientific community and paved the way for many creative ideas about how chaotic signals can be employed in communications. The problem of digital communication using chaos has been addressed in many doctoral theses [1,2], survey papers [3] and regular scientific articles. In the following paragraphs, authors will pay attention to the latest developments in spread spectrum (SS) communication systems that employ chaos.

There are several approaches how to employ chaos for SS communications. In most of the approaches, chaos is employed as a means for the masking or scrambling of the transmitted signal. Broadband, noise-like nature of the chaotic signals makes them very well suited to these tasks. To decipher the transmitted information, it is necessary to employ synchronization techniques or apply differential encoding methods.

One of the most straightforward and widely-used approaches is to employ discrete chaotic sequences for spreading information bits over a longer time interval and wider

frequency range by multiplying information symbols by chaotic sequences. This technique, called direct sequence spread spectrum (DSSS), allows to decrease SNR of the received signal at the expense of frequency bandwidth. This, in turn, allows for maintaining communication over longer distances. Moreover, it is possible to use spreading sequences for multiple access. However, practical implementation of coherent chaotic DSSS requires complex and computation-demanding synchronization loops [4] for code acquisition [5,6] and tracking. It is worth mentioning, that in vast chaotic DSSS papers, the synchronization problem is skipped.

To overcome the synchronization issue, a non-coherent differential chaos shift keying (DCSK) communication scheme was proposed. In this very popular modulation scheme, the receiver does not need to synchronize code with the transmitter as the reference codes are transmitted along with the information-carrying codes, inherently wasting up to half of the available bandwidth. Recently, several mechanisms how for reducing synchronization overhead were proposed. In quadrature DCSK schemes [7–9], the reference signals get transmitted in the quadrature channel, whereas useful information is sent over in-phase (I) channel. Further advances in DCSK development employ Walsh codes [10], permutation matrices [11,12], unitary transforms [13], Hilbert transform [14], multi-carrier techniques [15], combining direct and time-reversed sequences [16]. There are researches devoted to the employment of DCSK [17] in conjunction with Reconfigurable Intelligent Surface (RIS) which has a great potential for terrestrial and space communications [18]. Many recent papers propose to combine DCSK and index modulations [19,20]

If chaotic spreading is applied in the frequency domain, the resulting spread spectrum modulation is referred to as chaotic FHSS [21]. It is worth mentioning that DSSS can be easily converted into FHSS by applying frequency modulation (FM) [22]. In our research, we also use this technique.

Spread spectrum techniques can employ not only discrete-time sequences but also continuous-time waveforms. For example, chaotic spreading sequence (CSS) modulation technique [23], which is employed in the LoRa PHY, can be considered as frequency domain spreading using sawtooth waveforms. Moreover, in a recently published highly-cited paper [24], it was shown that nonlinear chirps provide even better performance compared to CSS. There are few papers [25] addressing the employment of continuous-time chaos for spreading, as there is a huge challenge with the implementation of spreading waveform synchronization. If the spreading waveform is purely chaotic, correlation-based techniques can not be employed.

In [26], authors proposed to use chaotic synchronization phenomena [27] for continuous time "code acquisition". In fact, any chaos shift keying (CSK)-based communication system can be considered as SS system, as chaos potentially can provide "code gain" similarly to discrete-time SS systems. In the publication [28], authors have added FM on top of CSK to provide better control over spectrum bandwidth and provide extra robustness against CFO issues.

There is an interesting emerging research [29] which employs short continuous-time chaotic pulses, which are produced by the finite impulse response (FIR) filter with chaotic taps, which are controlled by the information sequence to be transmitted. On the receiving side, an adjustable bank of the matched filters with chaotic impulse responses is used as well. Considering SS nature, the proposed system demonstrates excellent results. Noticeably, synchronization for the predicting of the next symbol waveform can be implemented using artificial neural network (ANN) [30].

From the recent publications about the analog implementations of chaos-based communication systems, it is worth mentioning [31], where a completely analog system is studied [32], where chaos-based data transmission system, which employs analog chaos generators and STM32 microcontroller unit (MCU) for the data part [33], where the experimental study of analog frequency modulated chaos shift keying (FM-CSK)-based data transmission system with offline detection is employed.

Digital implementations of chaos-based communication systems mostly employ off-the-shelf SDRs hardware, such as Universal Software Radio Peripheral (USRP). In one of the earliest papers [34], J. Kaddoum et al. presented the implementation of DCSK in GNU

Radio. In [35], authors presented the implementation of chaotic asynchronous code division multiple access (CDMA) in GNU Radio and USRP. In publication [36], authors presented National Instruments PXI$^{TM}$-based implementation of frequency modulated differential chaos shift keying (FM-DCSK), where chaotic wavelets are frequency modulated and later keyed using DCSK. In publications [37,38], authors demonstrated implementation of chaotic coherent DSSS in LabView$^{TM}$ and USRP.

One of the issues with off-the-shelf SDRs is that they usually require an external computer to run the software part of the system. The computer, which is connected to the SDR by universal serial bus (USB) or Ethernet cable, represents a huge potential bottleneck, which limits the sample rate in the communication system. For the implementation of SS or ultra-wideband (UWB) systems, this can be a real issue as those modulation types require substantial bandwidth. In 2020 scientists from our research group started work on field-programmable gate array (FPGA)-based implementation of chaos-based antipodal chaos shift keying (ACSK) system [39], which potentially will allow using sample rates above 100 MHz.

This paper is organized in a classical manner. Section 2 outlines the motivation of the research, Section 3 is devoted to the principles and models of ACSK, and the overall structure of the FM-ACSK transceiver. Section 4 is focused on the practical implementation of chaos generators, chaotic code synchronization and symbol synchronization. Section 5 is devoted to the description of techniques used for the evaluation of the transceiver's performance. Finally, Sections 6 and 7 discuss results and summarizes the work.

## 2. Motivation for the Research

This paper is devoted to the practical implementation and field testing of FM-ACSK communication system, which employs continuous-time spreading for the modulation and chaotic synchronization for the despreading of the received information bits. Although the theme of the practical implementation of analog or digital chaos generators is revealed relatively well in the scientific literature, the practical implementation of communication systems is addressed scarcely. The given work is motivated by the necessity to explore the real-world performance of FM-ACSK systems, developed in previous works [26,28,39,40] of our research group. Moreover, the given contribution aims to prove the effectiveness and superiority of chaotic synchronization for the demodulation of chaotic spread-spectrum signals.

## 3. FM-ACSK System's Description

FM-ACSK communication system is a combination of FM and ACSK modulations. ACSK is based on chaotic synchronization between master and slave chaos generators. In this section, mathematical principles of chaotic signal generation and synchronization are described. The description of the two-step modulation, which was first published in [41], is also provided.

### 3.1. Chaos Generator and Synchronization

As a base of ACSK communication system, a fourth-order modified Chua's circuit chaos generator was used. This circuit can be described as a system of four differential equations and piecewise linear function:

$$\begin{cases} \dfrac{dp_1}{dt} = -g(p_1, p_3)(p_1 - p_3) - p_2 \\ \dfrac{dp_2}{dt} = p_1 + \gamma p_2 \\ \dfrac{dp_3}{dt} = \theta(g(p_1, p_3)(p_1 - p_3) - p_4) \\ \dfrac{dp_4}{dt} = \sigma p_3 \end{cases}, \tag{1}$$

$$g(p_1, p_3) = \begin{cases} c(p_1 - p_3 - d) & (p_1 - p_3) > d \\ 0 & (p_1 - p_3) \leq d \end{cases}, \tag{2}$$

where $p_1, p_2, p_3, p_4$ are system's state variables, $\gamma$, $\theta$, $\sigma$, $c$, $d$ are system coefficients and $g(p_1, p_3)$ is piecewise linear function. System coefficients characterize the chaotic behavior of the system.

Master chaos generator's output signal $R_{out}$ consists of the weighted sum of the system's state variables $p_n$ and piecewise function $g$:

$$R_{out} = k_1 p_1 + k_2 p_2 + k_3 p_3 + k_4 p_4 + g(p_1, p_3), \tag{3}$$

where weight coefficients $k_1$, $k_2$, $k_3$ and $k_4$ define the output signal form variation. Weight coefficients can be potentially used for multiple access implementation, by changing their values for each system. The initial system's state conditions and system coefficient values are shown in Table 1. These values ensure stable chaotic signal generation as provided in [41], and the sample of generated chaotic signal is shown in Figure 1. Equation (3) can be visualized as shown in Figure 2 in the master chaos generator part.

**Table 1.** Initial system's state conditions and system coefficients.

| $p_1$ | $p_2$ | $p_3$ | $p_4$ | $\gamma$ | $\theta$ | $\sigma$ | $c$ | $d$ | $k_1$ | $k_2$ | $k_3$ | $k_4$ |
|---|---|---|---|---|---|---|---|---|---|---|---|---|
| 0.05 | 0.06 | 0.07 | 0.08 | 0.5 | 10 | 1.5 | 3 | 1 | −2.6302 | −0.6054 | 0.587 | 0.7763 |

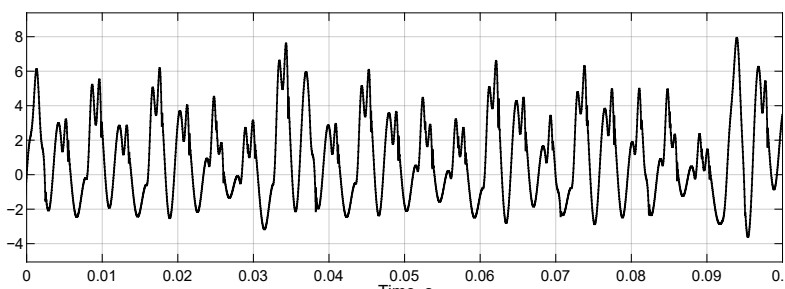

**Figure 1.** Master chaos generator's output signal $R_{out}$.

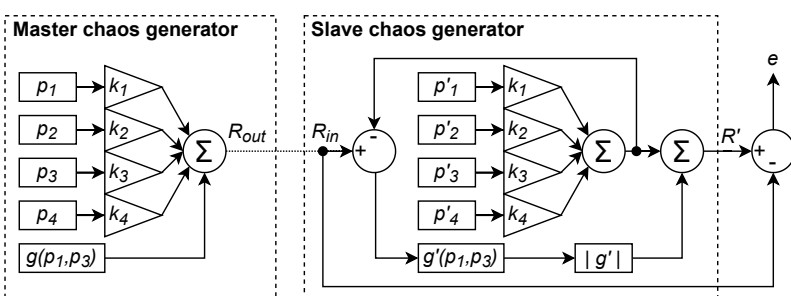

**Figure 2.** Block diagram of master and slave chaos generators.

Chaotic synchronization recreates the master chaos generator's state variables in the slave chaos generator, which is used to estimate received binary symbol values. Chaotic synchronization is carried out by the linear error feedback method. Both chaos generators have state variables $p_n$, $p'_n$ and equal weight coefficients $k_n$. The slave chaos generator's main difference compared to the master chaos generator is in the piecewise linear function that is being restored by subtracting the re-created weighted sum of the state variables $p'_n$ from the input signal $R_{in}$:

$$g'(p_1, p_3) = R_{in} - k_1 p'_1 + k_2 p'_2 + k_3 p'_3 + k_4 p'_4, \tag{4}$$

where $R_{in}$ is the input chaotic signal, $p'_1$, $p'_2$, $p'_3$, $p'_4$ are state variables of the slave chaos generator and $g'(p_1, p_3)$ is the restored piecewise linear function. Initial slave chaos generator's state conditions are shown in Table 2.

**Table 2.** Initial system's state conditions of slave chaos generator.

| $p'_1$ | $p'_2$ | $p'_3$ | $p'_4$ |
|---|---|---|---|
| 0.5 | 0.6 | 0.7 | 0.8 |

In the ideal communication system, $R_{in} = R_{out}$, but in a practical system, input chaotic signal $R_{in}$ is a sum of output signal $R_{out}$ and a combination of distortions $\eta$ from communication system's modulation, filters and transmission channel noise:

$$R_{in} = R_{out} + \eta ,\tag{5}$$

where $\eta$ describes all distortions between master and slave chaos generators, which is shown as a dotted line in Figure 2.

The slave chaos generator's output signal $R'$ is formed similarly as in the master chaos generator, but using an absolute value of recreated piecewise linear function $g'$:

$$R' = k_1 p'_1 + k_2 p'_2 + k_3 p'_3 + k_4 p'_4 + |g'(p_1, p_3)| .\tag{6}$$

Chaotic synchronization is established when master and slave state variables are equal, taking into account signal transmission delay. Piecewise linear function $g(p_1, p_3)$ values, according to (2), are always positive. The lack of chaotic synchronization can lead to the recovery of negative values from the function $g'$, which is incorrect. Thus, the absolute value of the function $g'(p_1, p_3)$ is used to estimate the synchronization error. To evaluate the synchronization error, the slave chaos generator's output signal $R'$ is subtracted from its input $R_{in}$:

$$e = R' - R_{in} ,\tag{7}$$

where $e$ is the synchronization error. This synchronization error is inconsistent due to the fact that the piecewise linear function at some moments is equal to zero. To ensure consistency in the estimation of the synchronization error, the root mean square (RMS) of it is calculated.

### 3.2. FM-ACSK Modulation

FM-ACSK communication system is made of two modulation layers—ACSK and FM modulations. As shown in Figure 3, FM is added on top of the ACSK modulation. In the ACSK layer, three chaos generators are used—one master chaos generator in the transmitter's ACSK modulator and two slave chaos generators in the receiver's ACSK demodulator. In the transmitter's ACSK modulator part, the master chaos generator's output $R_{out}$ gets passed through a switch, where $R_{out}$ gets inverted if the transmitted data bit is "1", or gets passed without a change if data bit "0" is transmitted. In the receiver's ACSK demodulator part, input signal $R_{in}$ is passed into two slave chaos generators—into one without signal inversion and into another one with inverted $R_{in}$ signal. This helps to determine if the transmitted chaotic signal was inverted or not in the transmitter by comparing two slave chaos generator outputs. At any given moment, one of the generators will provide the smallest estimated synchronization error compared to another generator. In order to decide the value of the transmitted data bit, RMS values of both slave chaos generators' synchronization errors are compared.

ACSK demodulator analyzes a relatively long interval of the chaotic signal to estimate the received data bit, as seen in Figure 4. Therefore, it can be said that the received signal's despreading in the ACSK demodulator is carried out by means of chaotic synchronization, which cardinally differs from the approaches used in classical SS systems. In classical SS systems, despreading is performed by means of correlation, which requires signifi-

cantly higher computation resources compared to chaotic synchronization. In [26], more information about ACSK can be found.

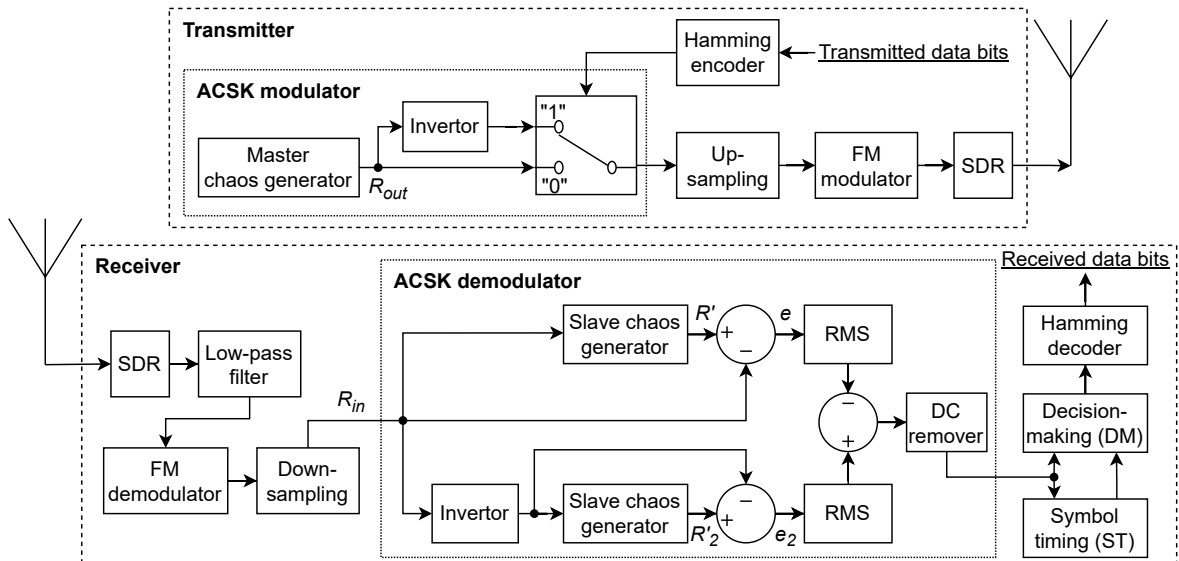

**Figure 3.** Block diagram of SDR-based FM-ACSK transceiver.

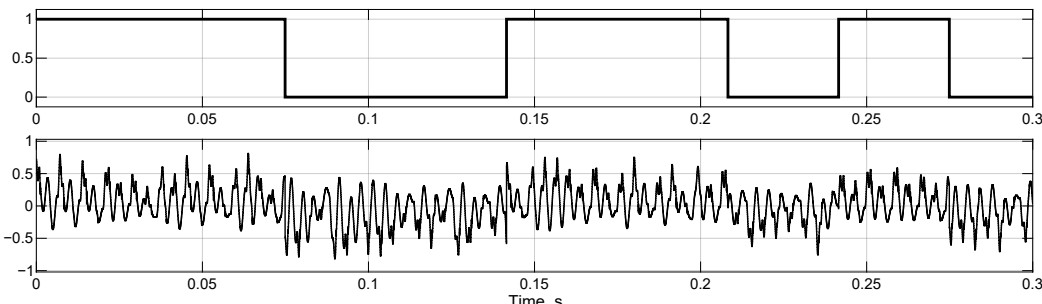

**Figure 4.** Spreading of the data bits using a continuous chaotic waveform. Transmitted data bits are drawn in the upper graph, switched chaotic waveform-lower graph. Each data bit is spread with approximately 9 oscillations of the chaotic waveform.

On top of ACSK modulation, FM is applied. The main purpose of FM is to spread transmitted chaotic signal even further, over the wider frequency range (see Figures 5 and 6), to diversify the signal and increase immunity to intentional or unintentional interference. Additionally, employment of non-coherent FM also increases immunity to CFO by lowering requirements for the stability of the carrier frequency. Additionally, the transmitted FM signal has a constant envelope, which allows using energy-efficient power amplifier (PA) with small backoff.

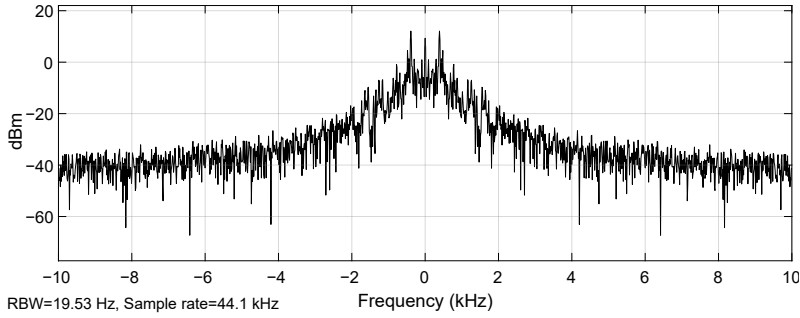

**Figure 5.** Baseband spectrum of the ACSK waveform (spread data bits).

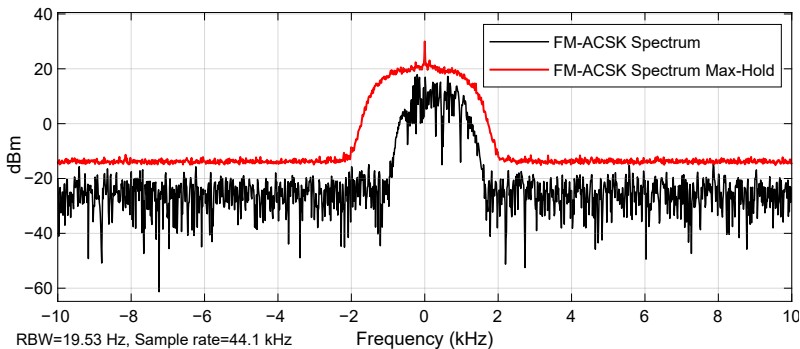

**Figure 6.** Baseband spectrum of the FM-ACSK at 10 kHz FM deviation.

## 4. SDR Implementation of FM-ACSK System Prototype

To test out and obtain results from the proposed communication system in real-world conditions, SDR-based prototype was created. This prototype's software part consists of a transmitter and receiver that are implemented from MATLAB Simulink models employed in our previous research [41]. The hardware consists of Adalm Pluto$^{TM}$ SDRs connected to one or two computers. The parameters of the tested system are shown in Table 3.

**Table 3.** Parameters of an FM-ACSK system prototype.

| Carrier Frequency | SDR Baseband Sample Rate | ACSK Sample Rate | Samples Per Bit | Data Rate | FM Frequency Deviation |
|---|---|---|---|---|---|
| 867 MHz | 240 kHz | 10 kHz | 1000 | 9.9 bit/s | 10 kHz |

### 4.1. FM-ACSK Transmitter

As shown in Figure 3, the transmitter of the prototype system consists of ACSK modulator with master chaos generators. This modulator's output is connected to upsampling block that converts ACSK modulator's sample rate into SDR's baseband sample rate. This upsampling makes it possible to use large FM deviations in the baseband FM modulator. Finally, the output of FM modulator is sent to Adalm Pluto$^{TM}$ SDR driver. The antipodal shift keying is performed by switching between inverted or direct master chaos generator's signal $R_{out}$ depending on the transmitted data bits. Transmitted data in the system with a Hamming encoder is modified by adding parity bits for single error correction capability.

### 4.2. Master Chaos Generator

In accordance with the description of the master chaos generator in Section 3.1, the generator in the transmitter's ACSK modulator is built. To be able to run this ACSK modulator with SDR, the Simulink model should run in discrete time. Therefore, a continuous-time model with differential equations must be converted into a discrete-time model with difference equations. The equations for the continuous-time "State-space" unit were:

$$\dot{x}_c = A_c x + B_c u = u \\ y = C_c x + D_c u = x$$ (8)

where $u$ is the input, $x$ is the state, $y$ is the output. As "state-space" units are integrators $A_c = 0$, $B_c = 1$, $C_c = 1$, $D_c = 0$. The respective "Discrete state-space" blocks are described by equations:

$$x(n+1) = Ax(n) + Bu(n) \\ y(n) = Cx(n) + Du(n)$$ (9)

where parameters $A$, $B$, $C$, $D$ can be determined using discretization technique [42]:

$$A = e^{A_c T}, \quad B = \int_0^T e^{A\tau} B_c d\tau, \quad C = C_c, \quad D = D_c,$$ (10)

where $T$ is the relative sample time which depends on the dynamics of the continuous system and it must be less than 0.5. After setting $T = 0.05$, we get $A = 1$, $B = T = 0.05$, $d = 1$, $D = 0$. Considering that value of $T$ is 10 times smaller than the critical value (0.5), the experimental evaluation has shown that discretized system behaved exactly as a continuous one.

Implemented in Simulink master chaos generator's scheme is depicted in Figure 7.

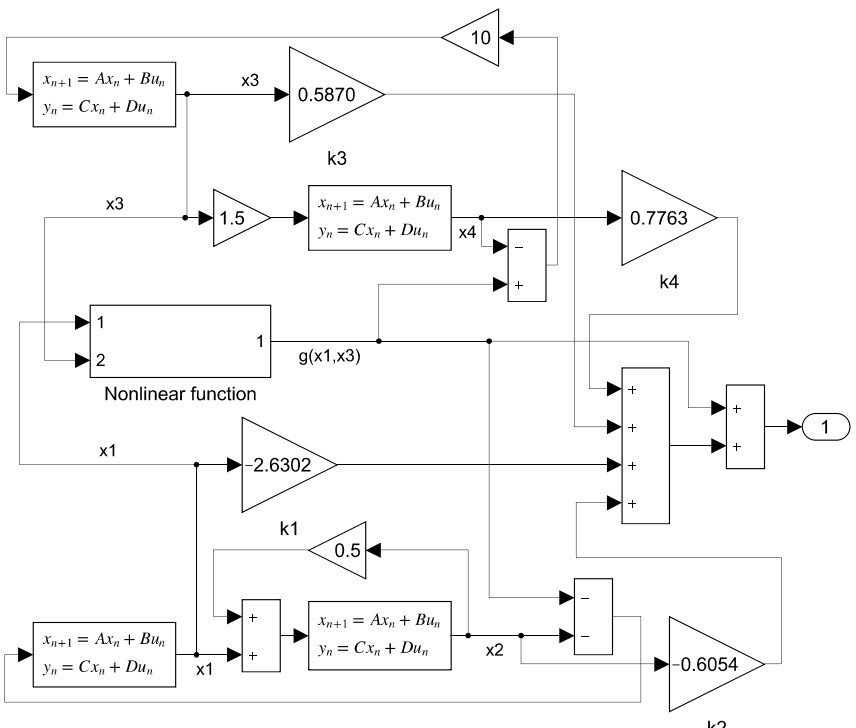

**Figure 7.** Implementation of chaotic generator in the ACSK modulator.

### 4.3. Hamming Code

Hamming codes are error-correcting codes that can detect and correct a single error in the encoded binary data string. These codes were first presented in [43]. In this prototype communication system, a (1023, 1013) Hamming code is used. This code has 1013 data bits and 10 parity bits with a total bit number of 1023, which comes from these two equations:

$$k = 2^m - m - 1, \tag{11}$$

$$n = 2^m - 1, \tag{12}$$

where $k$ is the number of data bits, $m$ is the number of parity bits and $n$ is the total bits after the addition of parity bits $m$ to the data. Lower $(n, k)$ Hamming codes provide better coverage of data bits, raising the probability of right error correction, at the expense of higher transmitted bit count for the same binary string's length, lowering transmission speed.

### 4.4. FM-ACSK Receiver

The communication system's receiver part consists of receiving SDR, low-pass filter (LPF), FM and ACSK demodulators as seen in Figure 3. Additionally, there are symbol timing synchronization and decision-making (DM) units. Before any demodulation is carried out, LPF is used to limit received noise when SDR's baseband rate is larger than the useful bandwidth. Downsampling, similar to upsampling in the transmitter, is used to decrease and align SDR's sample rate with that of ACSK demodulator, which is working at a lower sample rate. A Hamming decoder is added after DM unit to decode and correct an error if it is possible.

Two different types of DM units were tested. The first type is a classic strobe detector (SD) that, at the optimal time instant, samples waveform at the output of ACSK demodulator. The second type—energy detector (ED)—calculates the mean value of the whole symbol interval to make a decision.

### 4.5. Slave Chaos Generators

The receiver contains ACSK demodulator, which is made from two identical slave chaos generators. These generators can synchronize either to inverted or direct chaotic waveform $R_{in}$, looking at Figure 3 connections from the downsampling block to the generators. Slave chaos generators' internal structure is very similar to the transmitter's master generator's structure that was described in Section 4.2. The main difference between slave and master generators is in the nonlinear function's reconstruction from input signal $R_{in}$ that was described in Section 3.1. The design of the receiver's slave chaos generator implemented in Simulink is shown in Figure 8.

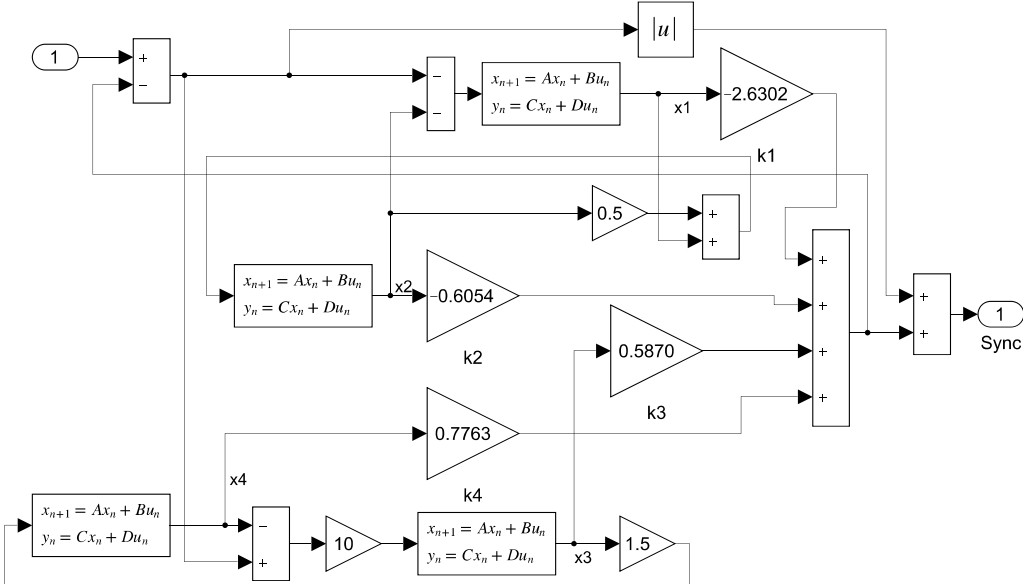

**Figure 8.** Implementation of the chaotic generator and synchronization circuit in the ACSK demodulator.

### 4.6. Symbol Timing Synchronization Unit

A timing synchronization unit is necessary for making decisions about the received symbols at optimal time instants. As was said before, the modem exploits chaotic synchronization for the despreading and demodulation of the encoded messages. In the case of the binary modulation, used in this research, there are two generators in the receiver and one of them is synchronized. At the output of RMS units in Figure 3, there is a gradually increasing signal when the respective slave chaos generator does not have chaotic synchronization. In contrast, when the generator is synchronized, its RMS unit's output is close to zero. The DM is carried out from the summary waveform, shown in Figure 9b, that is produced by the subtraction of signals from two RMS units. At the time instant when the summary waveform reaches the positive or negative peak, there is an optimal decision point for the best approximation of the received bit. The triangular waveform at the input of the DM unit, causes degradation of BER even at tiny symbol timing offset (STO). In terms of telecommunication terminology, it can be said that the eye diagram at the input of the DM unit is in the shape of a rhombus (see Figure 9a).

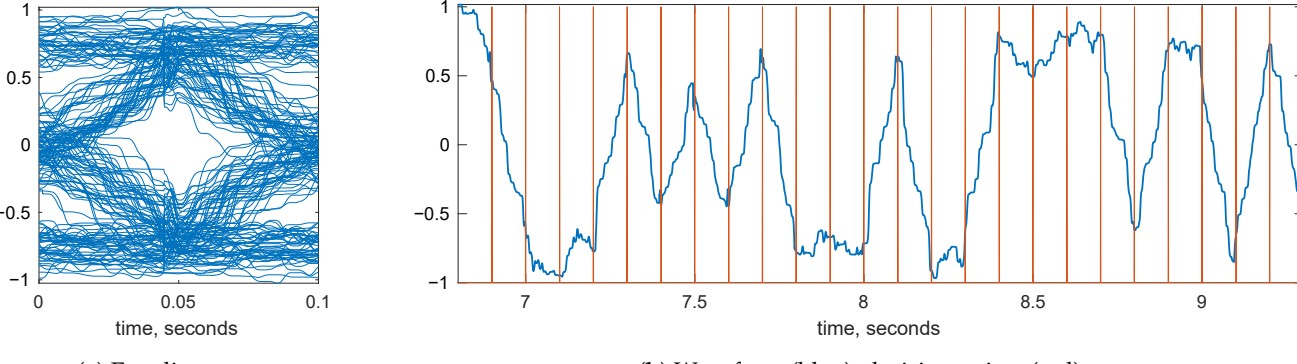

(**a**) Eye diagram

(**b**) Waveform (blue), decision points (red)

**Figure 9.** Signal at the input of decision-making unit.

For the estimation of the received bits, two alternative detectors were employed. The first one—SD—punctured the triangular signal at the input of DM at the time instants governed by symbol timing (ST) unit shown in Figure 9b. The second—ED—calculated the area of the waveform by summing all samples belonging to the symbol, i.e., around the decision point. The second detector was developed to mitigate the impact of the STO. The Simulink block scheme of ST unit is shown in Figure 10. The structure of ST unit resembles the classic loop synchronization scheme consisting of an offset estimator, loop filter and numerically controlled oscillator (NCO). The input signal is passed via the "Delay line" block which produces a series of windows shifted by one sample. "Find delay" finds the index of the maximum value in the particular window, which is selected by the symbol clock, and calculates offset from the midpoint. Error is sent to the loop filter which averages it over several symbol intervals and sends it to NCO which adjusts the clock to gradually decrease the offset.

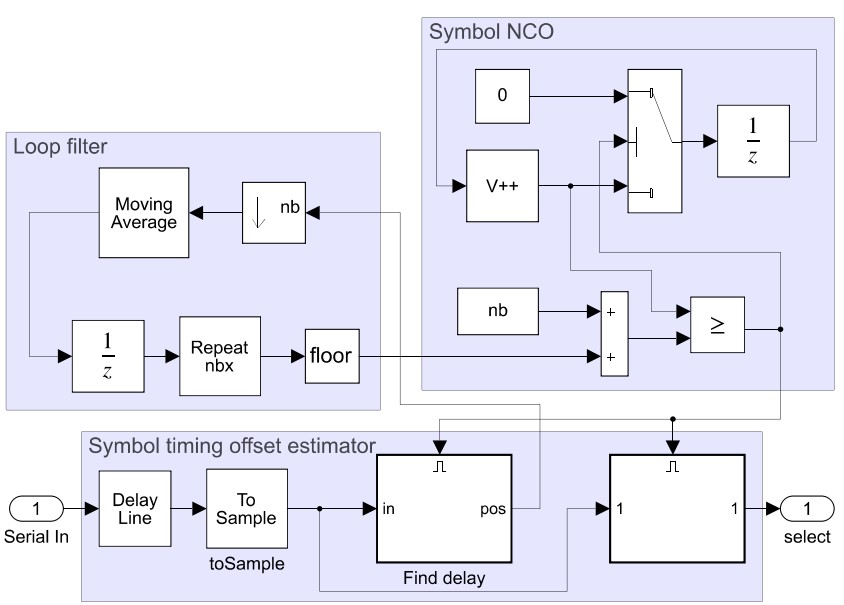

**Figure 10.** Implementation of the symbol timing (ST) unit.

### 4.7. Hamming Encoder

To understand how the employment of simple FEC affects the performance of FM-ACSK systems, simulation of FM-ACSK system baseband models (without SDR) has been carried out. Comparison of results from Simulink models with (1023, 1013) and (7, 4) Hamming FEC with different detectors is shown in Figure 11.

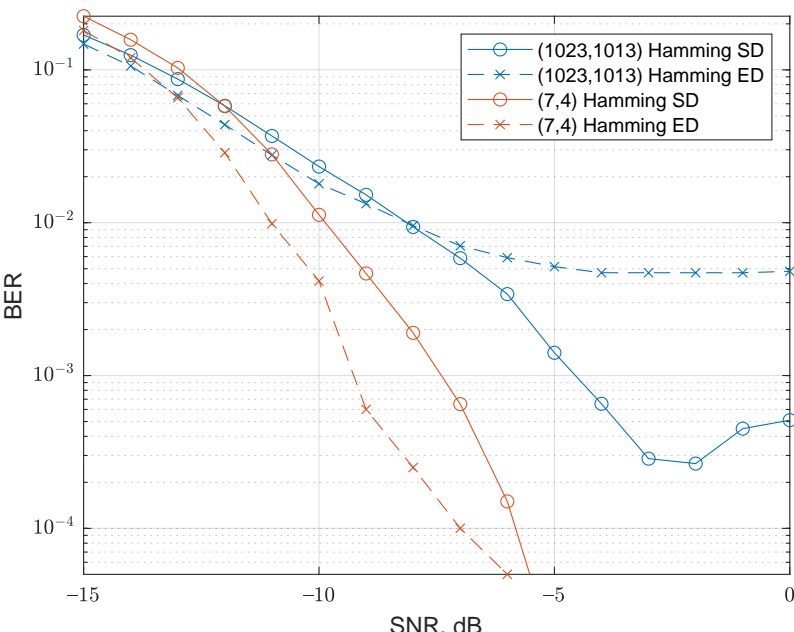

**Figure 11.** BER versus SNR for data transmission with (1023, 1013) and (7, 4) Hamming codes in FM-ACSK system. SD—strobe-based detector; ED—energy-based detector. FM deviation is 10 kHz.

As it can be seen, the system with a (7, 4) Hamming code provides better performance than the system with a (1023, 1013) Hamming code at SNR $>-11$ dB using SD. In the case of a (7, 4) Hamming code, the system with ED has an improvement in performance over the system using SD. System employing a (1023, 1013) Hamming code and ED has a slight performance improvement at SNR $<-8$ dB, but has a visible performance threshold at BER $\approx 4.7 \times 10^{-2}$.

## 5. Experimental Results

Experiments to measure the performance of SDR-based FM-ACSK prototype were carried out. In the experiments described in Sections 5.1–5.3 single Simulink model performed transmitter and receiver real-time simulations, running two or three SDRs connected to the same computer. Identical 27 cm long coil whip antennas were used for transmission and reception in these experiments. Antennas were located 4 m from one another. In the last experiment, described in Section 5.4, two completely independent computers with their own SDR units and antennas located 20 m apart from each other were used.

### 5.1. BER versus Transmit Power

In this test, the measurement of BER versus transmit power was conducted. In all measurements receiver's power was kept at 10 dB while the transmitter's power was varied from $-36$ dBm to $-16$ dBm. From the results that are depicted in Figure 12, it can be seen that, in all FM deviation and detector combinations, threshold effect is present, which could be related to the use of FM. Smaller FM deviation provides better communication quality with lower BER in both detector types. The system with an ED allows for achieving much lower BER than one with SD.

Figure 13 contains power spectra obtained from the signal logged immediately after the Adalm Pluto block in the receiver. These power spectra allow approximately estimate SNR of the receiver at transmit power $-26$ dBm and receiver gain 10 dB. This, in turn, allows recalculating the horizontal axis of Figure 12 into SNR using the approximate formula:

$$SNR = G_{\mathrm{s}} - G_{\mathrm{n}} + 10\log_{10}(K_{\mathrm{s}}) - 10\log_{10}(K_{\mathrm{n}}),\qquad(13)$$

where $G$ is power spectral density reading from the power spectrum and $K$ is a count of frequency bins occupied by the signal (subscript s) or noise (subscript n). For example,

in Figure 13 for spectrum with deviation 100 kHz, $G_s \approx -85$, $G_n \approx -95$, $K_s \approx 3$, $K_n \approx 2$. Conversely, SNR at transmit power level −26 dBm is approximately $-85 + 95 + 5 - 3 = 12$ dB. Therefore, to approximately covert the horizontal axis of Figure 12 into SNR, it is necessary to add 38.

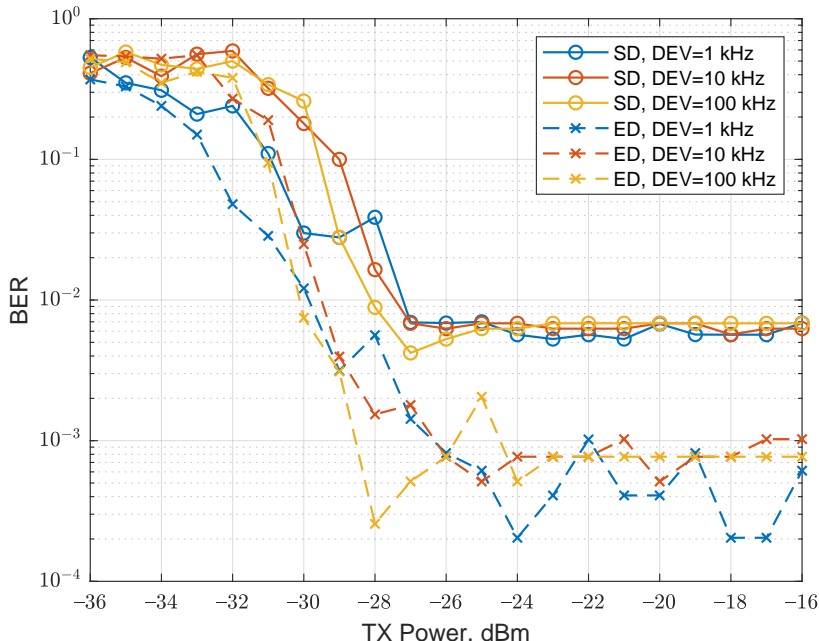

**Figure 12.** BER versus transmit power in dBm with different bit detectors in a single model scenario. SD—strobe-based detector; ED—energy-based detector; DEV—FM deviation. Receiver gain is 10 dB.

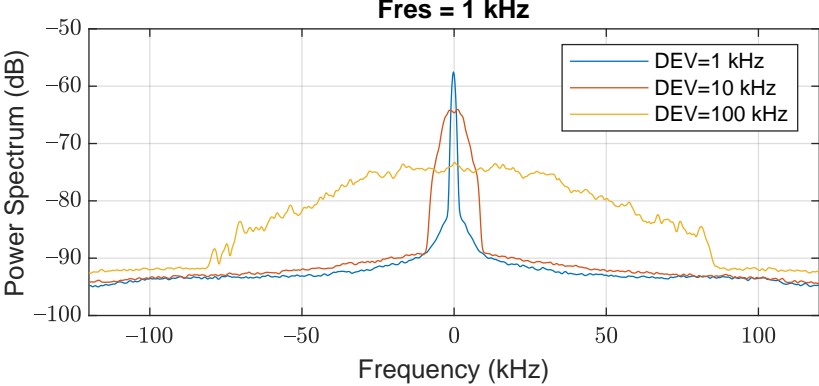

**Figure 13.** Measured power spectra at the output of Adalm Pluto. DEV—FM deviation. The transmitted power of Adalm Pluto is −26 dBm, and the receiver gain is 10 dB.

### 5.2. BER versus CFO

In this experiment, CFO was varied to measure its impact on the quality of communication. To measure it, the receiver's center frequency was detuned from the transmitter's carrier frequency, counting BER for each tested CFO value. Absolute CFO in Hz is calculated by multiplying FM deviation by relative CFO. The results of this experiment are depicted in Figure 14. These results show that the system is resistant to CFO of at least half of the FM deviation. The low-pass filter may impact the shape of the resulting curves, and a broader filter could improve the system's resistance to CFO at the cost of reduced SNR.

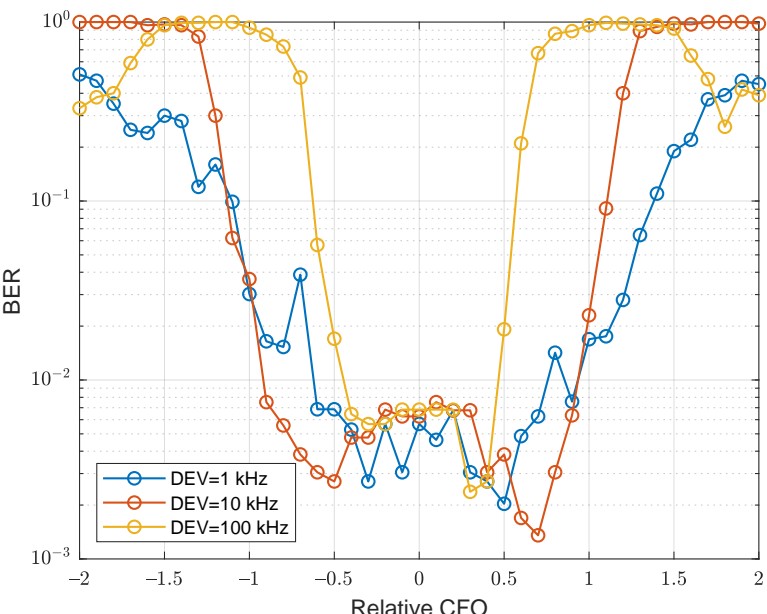

**Figure 14.** BER versus relative CFO with the strobe-based detector in the single model scenario. DEV—FM deviation. The receiver gain is 10 dB, and the transmitter gain is −26 dBm.

### 5.3. Multiple Access Interference

In this experiment, two FM-ACSK SDR transmitters and one FM-ACSK SDR receiver were used. Both transmissions were carried out at the same center frequency of 867 MHz using 27 cm long coil whip antennas. The distance between the transmitting antennas and receiving antenna was 1 m. The second transmitter's master chaos generator's coefficients, provided in Table 4, were changed to make synchronization between the second transmitter and receiver impossible. In the tests, the first transmitter's and receiver's power were kept constant while the second transmitter's power was varied. The result of this experiment is depicted in Figure 15. From the plot, it can be seen that a second transmitter on the same carrier frequency has a devastating impact on the quality of the communication. Therefore, it can be concluded that the given modulation scheme does not provide good multiple access opportunities out-of-the-box.

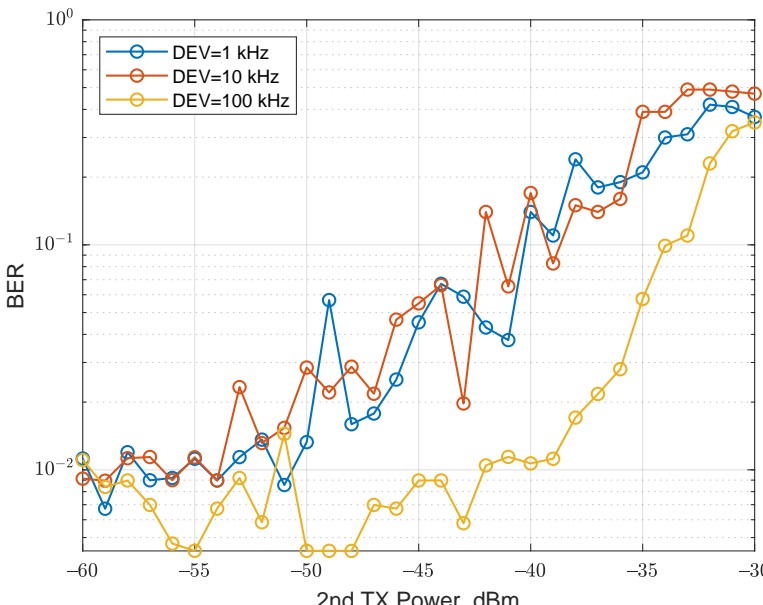

**Figure 15.** BER versus transmit power of the malicious transmitter. DEV—FM deviation. The receiver gain is 10 dB and the first transmitter power is −36 dBm. The receiver uses a strobe-based detector.

**Table 4.** Chaos generator's coefficients of the second transmitter.

| $\gamma$ | $\theta$ | $\sigma$ | $c$ | $d$ | $k_1$ | $k_2$ | $k_3$ | $k_4$ |
|---|---|---|---|---|---|---|---|---|
| 0.25 | 5 | 1.5 | 3 | 1 | 2.6302 | 0.6054 | −0.587 | −0.7763 |

### 5.4. Image Transmission between Two Computers

For evaluation of the practical application of FM-ACSK transceiver for wireless communication using data exchange protocols, image transfer tests between two independent computers were made. Image is transferred using a very simple proprietary protocol, described in textbook [44]. MATLAB scripts generate, receive and reconstruct data based on the implemented protocol. Additional information on how image frame generation is carried out is described in Appendix A and image reconstruction is described in Appendix B.

For the testing, the 8-bit greyscale image of size $100 \times 100$ pixels was used. This image contains 800 bits per pixel column which are transmitted in each frame. Each pixel has a value from 0 to 255 that defines its color on the greyscale. Transmission of an image was carried out between two independent computers, each connected to an Adalm Pluto$^{TM}$ SDR. The distance between sites was around 20 m. In both tests, the receiver's sensitivity was set to 30 dB and the transmitter's power to 4 dBm. Image transmission was carried out with and without the use of a (1023, 1013) Hamming code, and in both tests energy-based detector was used. Transmitted and received images are shown in Figure 16.

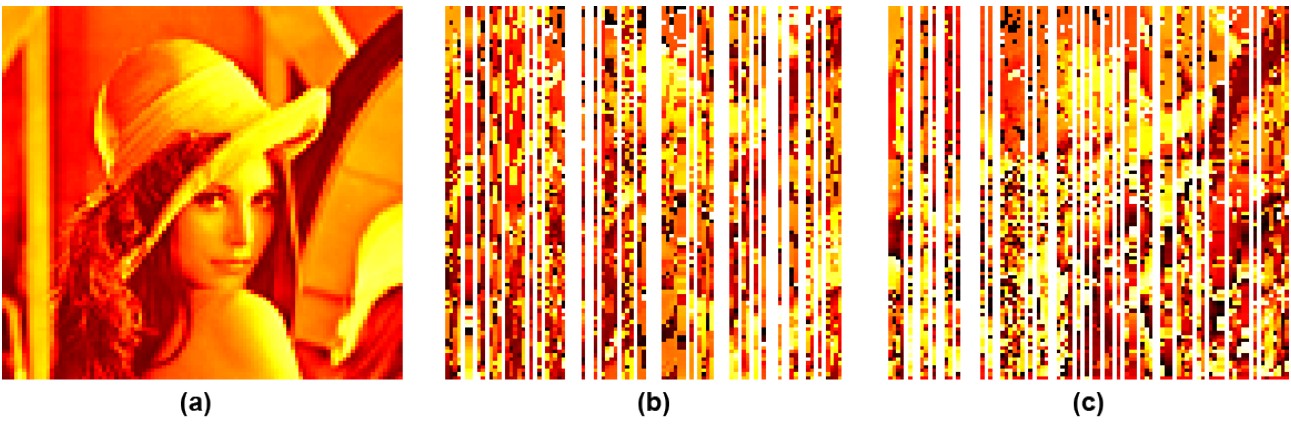

**(a)**　　　　　　　　　　　　　　　　　**(b)**　　　　　　　　　　　　　　　　　**(c)**

**Figure 16.** Received images after transmission between two computers. (**a**) Transmitted image. (**b**) Received image using FM-ACSK without a Hamming code. (**c**) Received image using FM-ACSK with a Hamming code.

Both received images have blank columns that were lost or moved to another column when the protocol header is damaged, and an incorrect sequence number in the header was received. Both images also have incorrectly colored pixels, which can happen due to wrong data bit reception and/or DM.

### 5.5. Comparison of FM-ACSK and FHSS

To compare the proposed modulation scheme with well-known modulation formats, baseband FHSS and FM-ACSK systems with similar parameters employing an additive white Gaussian noise (AWGN) communication channel were simulated. As can be seen from Figure 4, FM-ACSK employs approximately nine chaotic oscillations per one symbol, hence, the spreading factor is approximately nine. A similar FHSS system which employs a 9-chip long pseudo-noise (PN) sequence (1 1 1 1 −1 −1 −1 1 −1) was created.

FHSS system, built in accordance with scheme [45], spreads data bits by multiplying each bit with a bipolar spreading sequence. This signal is then up-converted and passed through FM, similar to FM part in FM-ACSK system. Despreading is carried out by correlating the received signal with a local synchronized spreading sequence. The resulting

waveform is then integrated and detected using a threshold detector. Auto-correlation function of the employed PN sequence is shown in Figure 17.

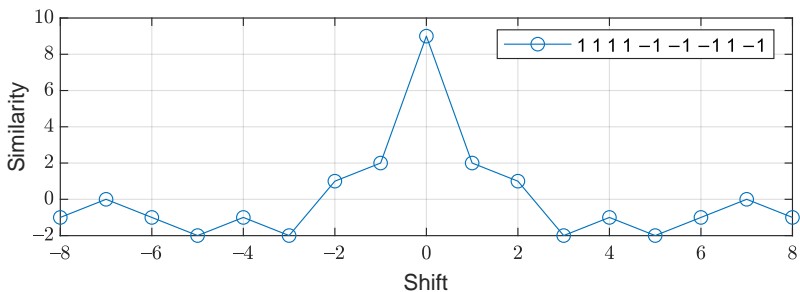

**Figure 17.** Auto-correlation of used spreading sequence.

BER plots obtained from baseband FM-ACSK and FHSS communication systems are shown in Figure 18. In this test identical bit patterns were sent over AWGN channel with variable SNR. FM-ACSK system was tested with both SD and ED detectors.

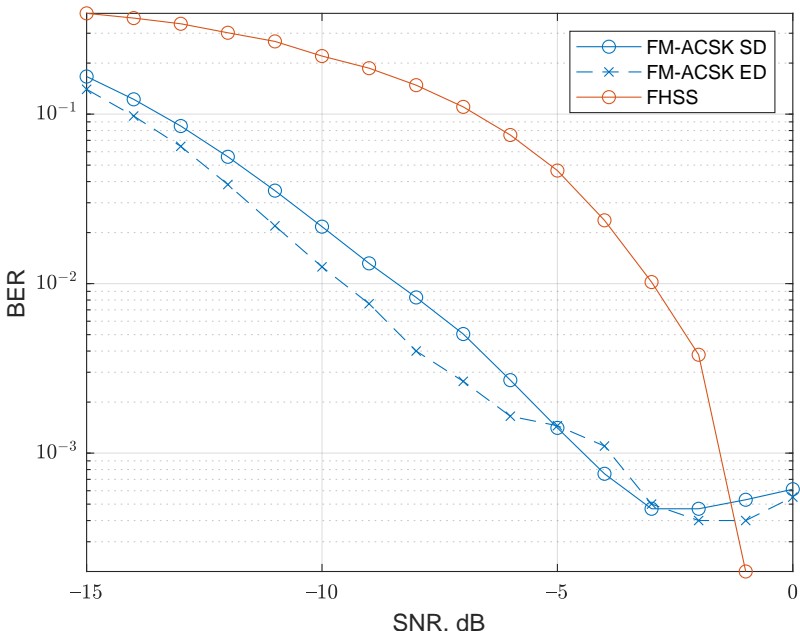

**Figure 18.** BER versus SNR for data transmission with FM-ACSK and FHSS. SD—strobe-based detector; ED—energy-based detector. FM deviation is 10 kHz.

Using both detectors, FM-ACSK system provides significantly better performance at low SNR values of $< -1$ dB. Employment of ED over SD slightly improves FM-ACSK system's performance. However, due to clearly observable error floor of FM-ACSK, at high SNR, correlation-based FHSS starts to outperform FM-ACSK.

## 6. Discussion

This research studies the implementation of FM-ACSK transceiver prototype in SDR. The purpose of the given research is a validation of the theoretical results obtained in earlier research and assessment of the real-world performance of a given modulation scheme taking into account practical problems, such as timing and frequency synchronization, multi-path propagation and multiple access interference (MAI). In order to make such a study, it was necessary to develop a new symbol synchronization unit that fits the particular scheme of ACSK demodulation. Moreover, compatibility of FM-ACSK with simple FEC schemes were tested. The experimentation and measurement campaign involved four different tests, which allowed us to learn about the real-world performance of FM-ACSK.

The BER versus transmit power measurement results have shown that the communication system is capable to provide acceptable BER $10^{-2}$ at SNR down to 8–10 dB. The newly proposed ED provides approximately 10 times lower BER compared to the SD one, proposed in earlier publications. The communication system is susceptible to STO due to the triangular shape of symbols at the input of the DM unit.

The BER versus CFO tests have shown that the transceiver is tolerant to large carrier frequency offsets and can keep BER $10^{-2}$ with CFO up to at least half of FM deviation. This property is achieved by the employment of non-coherent FM on top of coherent ACSK. Immunity to the CFO comes at the cost of a lower link budget as FM has a pronounced threshold effect and does not work at low SNR.

Unfortunately, MAI test has shown that orthogonality among FM-ACSK users, which employ similar chaotic oscillators with slightly different parameters, is low and there is considerable interference between users with the same carrier frequency. For providing multiple access in FM-ACSK based communication system, it is necessary to employ dedicated multiple access schemes such as frequency division multiple access (FDMA) or use advanced techniques such as non-orthogonal multiple access (NOMA) and successive interference cancellation (SIC).

The image transfer test has shown that the developed transceiver prototype can work fully autonomously and can be used for real data communication. Test results demonstrated acceptable performance of the transceiver in an office environment at the significant distance between the transmitter and receiver. Employment of Hamming FEC can provide significant improvement of BER, however, in case of insufficient code rate, FEC can cause the increase of BER.

From BER plots in Figures 12, 14 and 18 it can be concluded that FM-ACSK systems without FEC have error floor at approximately BER $\approx 10^{-3}$. This error floor can be caused by inaccuracies of the chaotic synchronization employed for the despreading of the chaotic waveforms.

## 7. Conclusions

The major scientific contribution of the given research work is in the multiple experimental results which provided real-world validation of FM-ACSK modulation scheme. Moreover, a novel symbol timing synchronization scheme and novel symbol detector were proposed.

The experiments have shown that the developed Adalm Pluto SDR-based FM-ACSK transceiver can provide real data communication with a speed of about 10 bps over a distance of 20 m using average transmit power about 3 mW. Further improvement of performance at low SNR can be achieved by boosting the processing gain via increasing the spreading factor of the spread-spectrum modulator. In accordance with Figure 4, the current implementation uses a relatively low spreading factor as there are just approximately 9 chaotic oscillations per one data symbol. However, a further increase in spreading factor can be achieved at the expense of data transfer speed or an increase of chaos generators' sample rate. Employment of FEC can decrease BER to arbitrary low values at the cost of the transmission speed.

The proposed solution suits very well wireless sensor network (WSN) applications which are characterized by relatively low transfer speed and low SNR due to low-power transmission or large distance. However, WSN applications require compact and cost-efficient transceivers due to large-scale deployments of the sensor nodes. Transitioning from the MATLAB-based prototype into a compact and cost-efficient solution can be carried out in several ways.

Firstly, SDR unit can be kept whereas the software part can be rebuilt using open-source tools such as GNU Radio and Python which can run on a single-board computer such as a Raspberry PI. This, along with the size and cost reduction, can provide also a performance boost.

If further reduction of cost and size is necessary, Simulink code can be converted into very high-Speed integrated circuit hardware description language (VHDL) using Simulink Coder[TM] [40] and run on FPGA connected to the radio frequency (RF) transceiver integrated circuit (IC), for example, AD9363, which used within Adalm Pluto. Compared to SDR unit-based solutions, FPGA can run significantly faster and provide much larger spreading factors or data transfer rates. Availability of Simulink Coder[TM] was one of the major motivations for the selection of MATLAB versus GNU Radio in our research.

For even larger cost and size reduction, VHDL code can be converted into application-specific integrated circuit (ASIC) layout and integrated with a custom RF part. Furthermore, the transceiver and especially chaos generators can be built partly analog, providing a significant reduction of power consumption.

**Author Contributions:** Conceptualization, A.A.; methodology, A.A.; software, N.T.; validation, A.A.; formal analysis, A.A.; investigation, A.A.; resources, A.A.; data curation, N.T.; writing—original draft preparation, N.T.; writing—review and editing, A.A.; visualization, N.T.; supervision, A.A.; project administration, A.A.; funding acquisition, A.A. All authors have read and agreed to the published version of the manuscript.

**Funding:** This work has been supported by the European Regional Development Fund within Activity 1.1.1.2 "Post-doctoral Research Aid" of the Specific Aid Objective 1.1.1 "To increase the research and innovative capacity of scientific institutions of Latvia and the ability to attract external financing, investing in human resources and infrastructure" of the Operational Programme "Growth and Employment" (No.1.1.1.2/VIAA/2/18/345).

**Conflicts of Interest:** The authors declare no conflict of interest.

## Appendix A

The transmitted frame is made of header and payload parts. The header part stores the information about the sent frame and consists of fixed PN preamble for the frame synchronization, frame sequence number, end frame flag, header, payload and padding lengths. The payload part is a combination of data and padding bits. Padding bits are used to fill the frame to a predefined bit length when the sum of header and data bits is smaller than the frame length. The structure and length of the transmitted frame in bits are shown in Table A1.

**Table A1.** Structure and length of the transmitted frame in bits.

| Header | | | | | | Payload | |
|---|---|---|---|---|---|---|---|
| Preamble | Sequence Number | End Flag | Header Length | Payload Length | Padding Length | Data | Padding |
| 40 | 8 | 1 | 8 | 10 | 9 | 800 | 137 |
| 1013 | | | | | | | |

The total length of a frame of 1013 bits was chosen to easily encode and decode it using a (1023, 1013) Hamming code.

The script checks if the preamble is of even length and if not, add 0 to the end of it to make its length even. Frame sequence number stores the information about the frame's number which is the image column's number. The end frame flag shows if the transmitted frame is the last one. Header, payload and padding lengths store the information about the lengths of these parts of the frame.

Data consists of image column pixels in binary form. To create the frame's data, the image is loaded and reshaped to a single column of data. Then it is converted to binary format from decimal format and after that reshaped into a row as seen in Figure A1. Conversion to binary format is carried out with most significant bit (MSB) on the left side.

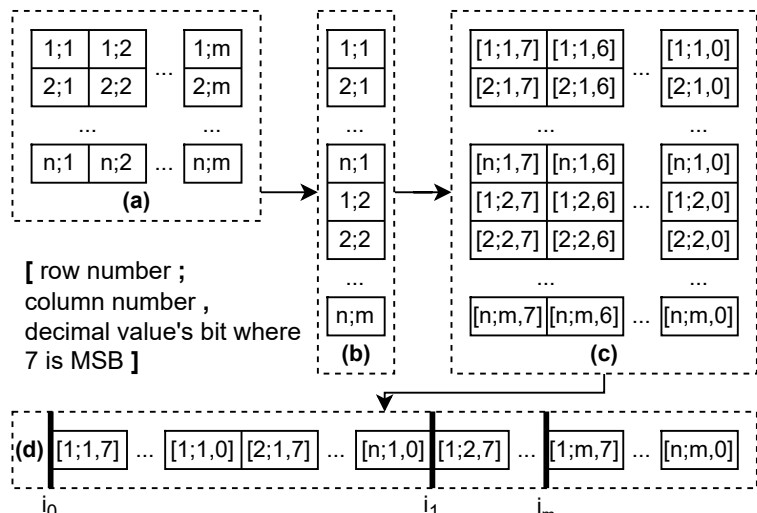

**Figure A1.** Creation of data bits from the image where *n* is image pixel row length, *m* is image pixel column length, and *i* is frame or image column number. (**a**) Image matrix with decimal values. (**b**) Image matrix reshaped into column vector with decimal values. (**c**) Matrix of each image pixel (matrix row) and its binary value in matrix columns. (**d**) Data row containing the image in bits where black lines show a start of an image pixel column.

Generated frame in binary format is sent as an input to the Simulink FM-ACSK transmitter system and transmitted via Pluto$^{TM}$ SDR.

### Appendix B

The image receiver uses a dedicated script for data extraction and collection from the frames. The same parameters for the preamble and lengths of a frame from Table A1 are used. FIR matched filter is used for the detection of PN preamble at the beginning of each frame. The synchronized frame from the receiver script's output is used as an input for the image reconstruction script. To receive and reconstruct an image, the transmitter's and receiver's protocol parameters for header and payload parts must be equal.

---

**Algorithm A1:** Frame sequence number lock algorithm

---

**Data:** $psn[4]$: previous sequence numbers, $sn$: current sequence number
**Result:** $snl$: sequence number lock, $sn\_ce$: sequence number count enable
**for** $i \leftarrow 4$ **to** $2$ **by** $-1$ **do**
  $\quad psn(i) \leftarrow psn(i-1);$           // Shift previous sequence numbers
**end**
$psn(1) \leftarrow sn;$                  // Save current sequence number
**if** $psn(4) + 1 = psn(3)$ **and** $psn(3) + 1 = psn(2)$ **and** $psn(2) + 1 = psn(1)$ **then**
  $\quad snl \leftarrow psn(1);$         // Set lock to the current sequence number
  $\quad sn\_ce \leftarrow 1;$           // Set sequence number count enable high
**end**
**if** $sn\_ce = 1$ **then**
  $\quad snl \leftarrow snl + 1;$        // Increment sequence number lock
**end**

---

---

**Algorithm A2:** Frame validity and out of bounds check

---

**Data:** *rsn*: received sequence number, *snl*: sequence number lock, *sn_ce*: sequence number count enable

**Result:** *valid*: frame validity

$valid \leftarrow 1$;

**if** $rsn < 1$ **or** $rsn > 100$ **then**

    **if** $sn\_ce = 1$ **then**

        $rsn \leftarrow snl$;          `// Received frame is after previous frame`

    **else**

        $valid \leftarrow 0$;          `// Received frame is not valid`

    **end**

**end**

---

**Algorithm A3:** Image column reconstruction

---

**Data:** *valid*: frame validity, *snl*: sequence number lock, *sns*[100]: sequence number storage, *rsn*: received sequence number, *rx*: received frame

**Result:** *image*: reconstructed image

$b\_l \leftarrow 8$;          `// Binary length of a pixel`

$header\_l \leftarrow 76$;          `// Header length`

$data\_l \leftarrow 800$;          `// Data length`

$column(1 \textbf{ to } 100) \leftarrow 0$;

**if** $valid = 1$ **and** $snl \geq 100$ **and** $sns(rsn) = 0$ **then**

    $data \leftarrow rx(header\_l + 1 \textbf{ to } header\_l + data\_l)$;

    **for** $i \leftarrow 1$ **to** $100$ **by** $1$ **do**

        $b\_data \leftarrow data((i-1) * b\_l + 1 \textbf{ to } i * b\_l)$;

        $column(i) \leftarrow (\text{decimal})b\_data'$;    `// Convert pixel from binary to`

        `decimal format`

    **end**

    $sns(rsn) \leftarrow 1$;

    $image(1 \textbf{ to } 100, sn) \leftarrow column$;          `// Add column to an image`

**end**

---

If header, payload and padding lengths are recovered with errors, these lengths are set to default values. After the conversion of header values, the validity of a received frame is checked. It is carried out by comparing the current received frame's sequence number and 3 previous sequence numbers which are described in Algorithm A1. If the frame is valid, the lock enable signal is set high and the next sequence number is locked. Lock enable signal is used in the case when the received frame's sequence number is out of bounds of acceptable sequence numbers, which is <1 and >100, which corresponds with the used image's width in pixels. When such a frame is received, and the lock enable signal is high, the received frame is assumed to be with a sequence number of a locked sequence number. If the lock enables signal is low, this frame is dropped and will not be recovered. The frame validation check is presented in Algorithm A2.

If the received frame is valid, its data content is converted from binary to decimal image column format, where the column's number is the frame's sequence number extracted from the frame's header. Conversion is carried out by reading 8 bits and saving them as decimal values, doing it for all 100 received pixels. If several frames with equal sequence numbers are received, only the first frame's data is saved. This image reconstruction is described in Algorithm A3.

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
