# Peer review of "Software-Defined Radio Implementation and Performance Evaluation of Frequency-Modulated Antipodal Chaos Shift Keying Communication System†"

_electronics, doi:10.3390/electronics12051240_

Round 1

Reviewer 1 Report

This work presents a wireless transmission system based on frequency modulation and antipodal chaos shift keying. This system is designed and implemented in a real scenario using software-defined radio and MATLAB Simulink. The Authors argue that the main advantage of this system lies in the fact that it is not necessary to use cross correlation for synchronization thus saving resources. The operation of the system and the experiments are clearly described. In some passages there are even too many details that could be omitted or moved to an appendix (section 4.4 Image Transfer) considering that the topics covered are marginal to the main topic of the article.

On the contrary, it would be worth investigating some themes that are central to the work:

1) is the transition from equations 1) and 2) to 7) and 8) - the discretization of the electrical signal for the implementation in Matlab Simulink - free of approximation effects?

2) the operation of the energy-based synchronization circuit described in fig. 9;

3) in the numerous performance graphs (i.e. BER vs SNR) in which some possibilities of the proposed system are compared, it would be advisable to also include one or more transmission systems known in the literature and which the Authors believe can be compared with the proposed one. This aspect is fundamental to evaluate the effectiveness of the proposed system.

4) is the passage from the simulated SDR+Maltlab Simulink environment to a real device immediate or are there hardware limits to be respected with particular reference to the realization of the Chua circuit?

Some text edit suggestions:

line 208: replace "-" with "is the"

line 222: delete "much" before "lower"

Figure 12 caption: replace "Transmit" with "Transmitted"

Figure 14 caption: replace "gain" with "power" before "is -36dBm"

line 397: "2" of "10-2" should be apex

line 397: replace "lave" with "half"

Reviewer 2 Report

This paper has significant theoretical work. The implementation contribution is still on the way. In the opinion of this reviewer Matlab Simulink and Pluto SDR are good and popular tools. However, their functions and scripts are far away from being a novel implementation. There are many similar contributions, nowadays, free software contributions and low-cost devices are more relevant and could gain more importance in the community.

This review asks the authors to add information about which kind of applications could be considered in BER 10-2 at SNR down to 8 – 10 dB, the data that are included in the discussion.

Round 2

Reviewer 3 Report

The authors have basically addressed my concerns, no further comment.